# MAKE IT NOISEASIER: BOOSTING TEXT-TO-VIDEO GENERATION WITH DIRECT NOISE OPTIMIZATION

## ABSTRACT

Diffusion models have significantly advanced text-to-video (T2V) generation, yet they still struggle with complex prompts involving intricate object interactions and precise attribute binding. While reward-based fine-tuning can improve compositional alignment, it is computationally costly and prone to reward hacking. In this work, we propose **NoisEasier**, a test-time training framework that improves T2V generation by directly optimizing latent noise with differentiable rewards during inference. To make this practical, we leverage fast video consistency models, enabling full gradient backpropagation within just 4 denoising steps. To mitigate reward hacking, we integrate multiple reward objectives that balance semantic alignment and motion quality, and propose a novel negative-aware reward calibration strategy that uses LLM-generated distractors to provide fine-grained compositional feedback. Experiments on VBench and T2V-CompBench show that NoisEasier consistently improves strong baselines, achieving over 10% gains in several dimensions and even surpassing commercial models like Gen-3 and Kling. Notably, these improvements are achieved within 25 optimization steps, requiring only 45 seconds per sample on two RTX 6000 Ada GPUs. Under the same wall-clock time budget, NoisEasier achieves human preference win rates exceeding CogVideoX-2B by 18.8% and Wan2.1-1.3B by 6.8%, demonstrating a competitive trade-off between performance and efficiency.

## 1 INTRODUCTION

Diffusion models (Sohl-Dickstein et al., 2015; Ho et al., 2020; Song et al., 2020b; Karras et al., 2022) have emerged as the state-of-the-art framework for both image and video generation, demonstrating impressive capabilities in producing high-fidelity, diverse, and temporally coherent content. Recent advances in text-to-video (T2V) generation (Guo et al., 2023; Chen et al., 2024a; Qing et al., 2024; Wang et al., 2025a) have made significant strides toward translating natural language descriptions into plausible frame sequences. Despite these successes, current models still struggle with faithfully rendering intricate compositional prompts (Huang et al., 2023; Tian et al., 2024; Sun et al., 2024). Challenges such as inaccurate object interactions, inconsistent attribute binding, and the generation of incoherent frames remain prevalent.

To address these issues, recent works draw inspiration from Reinforcement Learning from Human Feedback (RLHF) (Christiano et al., 2017; Ouyang et al., 2022) in large language models (LLMs), adapting similar strategies to align T2V models with human preferences. Early efforts like InstructVideo (Yuan et al., 2024) and VADER (Prabhudesai et al., 2024) fine-tune the diffusion model with differentiable rewards (Kirstain et al., 2023; Xu et al., 2023; Wu et al., 2023b), while later studies build human preference datasets for direct preference optimization (Liu et al., 2024; Ding et al., 2025; Cheng et al., 2025) or reward-based fine-tuning (Wu et al., 2024; He et al., 2024; Liu et al., 2025). Despite their promise, fine-tuned models are inherently fixed, limiting their controllability and adaptability to user-specific requirements.

Alternatively, another line of work investigates optimizing the latent noise during inference to avoid the limitations of model fine-tuning. Recent studies (Li et al., 2024d; Zhou et al., 2024; Qi et al., 2024; Eyring et al., 2024; Xu et al., 2025) have shown that initial random noise strongly influences the fidelity and semantic alignment of generated images, with optimized noise often yielding higher-quality results. Compared to model fine-tuning, noise optimization offers a parameter-efficient and

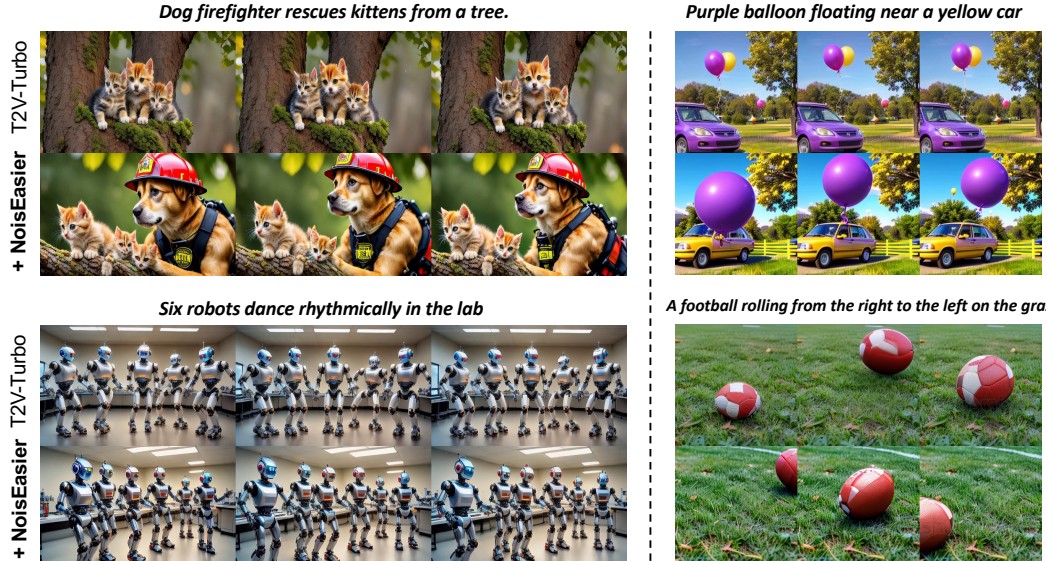

Figure 1: Qualitative results with and without NoisEasier across different prompts. **Left**: T2V-Turbo (VC2), 4-step generation; **Right**: T2V-Turbo (MS), 4-step generation.

flexible alternative, as it requires no weight updates, can readily adapt to new tasks or reward signals at test time, and has been successfully applied in image (Guo et al., 2024; Eyring et al., 2024), motion (Karunratanakul et al., 2024), and even music generation (Novack et al., 2024a). However, direct noise optimization for T2V generation remains underexplored due to two substantial challenges: (1) synthesizing a single video already takes tens of seconds to minutes, making iterative optimization impractical under realistic time and computational budgets; and (2) there is a lack of reward models trained on video-level preferences. While some recent works (He et al., 2024; Liu et al., 2025; Wang et al., 2025b) adapt multimodal large language models (MLLMs) with video preference data for model fine-tuning, their massive architectures make gradient tracking across denoising steps prohibitively expensive, discouraging their use in test-time optimization.

To address these challenges, we propose **NoisEasier**, a test-time training (Prabhudesai et al., 2023b; Gao et al., 2023; Kim et al., 2025) approach that refines the latent noise during inference using differentiable rewards. Instead of relying on large pretrained VDMs, we employ distilled video consistency models (Wang et al., 2023b; Li et al., 2024b) that enable efficient sampling within only 4 to 8 denoising steps, substantially reducing the computational overhead and making gradient-based noise optimization feasible. In the absence of video reward models, a naive solution is to utilize image-level reward models (Kirstain et al., 2023; Xu et al., 2023; Wu et al., 2023b). However, aggressively optimizing for such rewards will degrade video fidelity, especially motion dynamics and temporal coherence. Therefore, we integrate multiple reward objectives that balance diverse aspects of generation quality and introduce a motion reward objective to encourage temporal dynamics and smoothness. In addition, while these reward models can capture high-level human preferences, they provide limited compositional feedback, as they are primarily trained via pairwise ranking and often behave like bag-of-words classifiers (Yuksekgonul et al., 2022; Madasu & Lal, 2024). To this end, we propose a *negative-aware reward calibration* strategy that encourages alignment with the target prompt while distinguishing it from syntactically similar but misleading negative prompts, which shows significant improvements on attribute binding, spatial relationship, and numeracy.

With all reward models enabled, our method performs 25-step noise optimization in under one minute per sample on two RTX 6000 Ada GPUs. Extensive experiments on two widely used benchmarks, VBench (Huang et al., 2024) and T2V-CompBench (Sun et al., 2024), show consistent improvements over baseline models, with gains exceeding 10% across multiple metrics and even surpassing commercial systems such as Pika (Pika Team, 2024) and Kling (Kuaishou, 2024). These results underscore the importance of noise initialization in video generation and establish noise optimization as a practical and scalable alternative to full model fine-tuning.

## 2 RELATED WORK

**Video Generation with Human Feedback.** Reinforcement Learning from Human Feedback (RLHF) (Christiano et al., 2017; Ouyang et al., 2022) has shown great success in aligning large language models (LLMs) with human intent, motivating its extension to image and video generation. Existing work typically follows two paradigms. One is *preference-driven optimization*, where models are trained directly on large-scale human preference datasets. This includes policy gradient methods such as DDPO (Black et al., 2023) and DPOK (Fan et al., 2023), as well as direct preference optimization methods (Liang et al., 2024; Wallace et al., 2024; Yang et al., 2024a; Liu et al., 2024) that bypass explicit reward models. While effective, these approaches require large-scale preference annotations and cannot easily incorporate structured human knowledge. The other is *reward-driven optimization*, where pretrained reward models provide differentiable feedback for fine-tuning generative models (Yuan et al., 2024; Prabhudesai et al., 2024) or guide distillation for faster inference (Li et al., 2024a;b;c). Although these methods reduce training cost and improve stability, they are generally one-shot systems that lack mechanisms for iterative refinement. In contrast, we directly optimize latent noise via differentiable rewards, amortizing test-time refinement into training and enabling controllable, iterative text-to-video generation.

**Direct Noise Optimization.** Recent studies (Li et al., 2024d; Zhou et al., 2024; Xu et al., 2025) show that random noise strongly influences generation quality, with certain *golden noise* producing outputs more faithful and aesthetically pleasing to human preferences. DOODL (Wallace et al., 2023) first optimized initial noise in an end-to-end differentiable manner, improving classifier guidance and conditional generation. Follow-up efforts extended this idea to image editing (Chen et al., 2024b), compositional image generation (Guo et al., 2024; Tang et al., 2024), music (Novack et al., 2024b) and motion generation (Karunratanakul et al., 2024). While effective, most of these methods suffer from high latency during inference time, with DOODL (Wallace et al., 2023) and D-Flow (Ben-Hamu et al., 2024) requiring 10 to 40 minutes per sample. Recently, ReNO (Eyring et al., 2024) mitigates this by applying optimization to one-step diffusion models, reducing optimization time to under 20 to 50 seconds. Motivated by this, we propose direct noise optimization for video generation, where high computational cost and limited reward feedback present new challenges.

## 3 NOISEASIER: NOISE OPTIMIZATION WITH REWARD GRADIENTS

### 3.1 PROBLEM FORMULATION

Given a pretrained T2V diffusion model $\mathcal{G}_\theta$, the generation process begins from an initial latent noise vector $\mathbf{z}_T \sim \mathcal{N}(0, \mathbf{I})$ and progressively denoises it to synthesize a video sequence conditioned on a text prompt $c$. Our goal is to optimize the latent noise at test time under a differentiable reward function $\mathcal{R}$:

$$\mathbf{z}_T^* = \arg\max_{\mathbf{z}_T} \mathcal{R}(\mathcal{G}_\theta(\mathbf{z}_T, c)). \tag{1}$$

Here, $\mathcal{R}$ should measure the semantic alignment or perceptual quality of the generated video, and $\mathbf{z}_T$ is optimized to steer the denoising process toward higher-quality results. However, backpropagating gradients through the entire denoising trajectory of classic diffusion models (Ho et al., 2020; Song et al., 2020b) is non-trivial due to the large number of iterative steps. Recent approaches (Clark et al., 2023; Prabhudesai et al., 2023a; 2024) address the issue by skipping early steps and truncating gradients, but this sacrifices complete gradient flow and remains computationally expensive. In contrast, *video consistency models* (Wang et al., 2023b; 2024; Zhang et al., 2024; Li et al., 2024b) distill diffusion models into efficient generators that bypass hundreds of iterative steps by directly mapping noised inputs to clean outputs in just a few steps. This reduction makes full gradient backpropagation feasible. Building on this advantage, we adopt T2V-Turbo (Li et al., 2024b) as our backbone to enable practical noise optimization. Please refer to Section A.1 for more details.

Formally, the generation process in diffusion models can be viewed as solving a reverse-time SDE or its deterministic ODE counterpart over discretized timesteps. DDIM (Song et al., 2020a) provides a parameterized sampler that interpolates between these regimes via a hyperparameter $\eta$, where $\eta = 0$ yields a deterministic trajectory and $\eta > 0$ introduces stochasticity by adding Gaussian noise at each step. T2V-Turbo adopts the stochastic variant of DDIM by default, injecting random

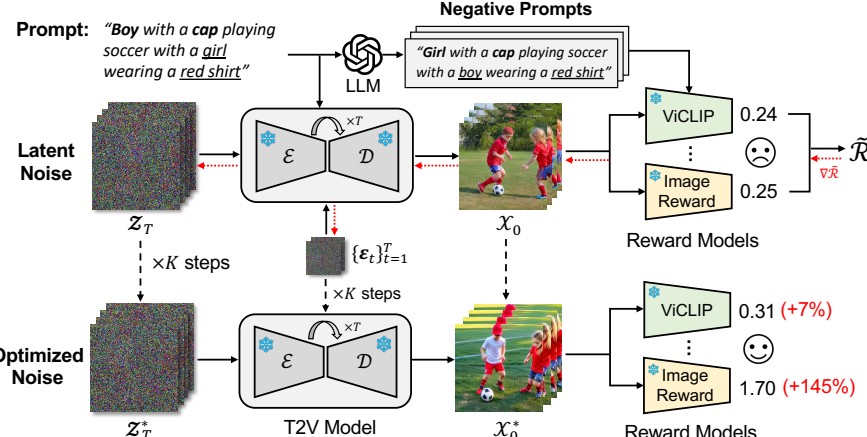

Figure 2: Overview of the proposed NoisEasier framework. Given a text prompt, both initial and intermediate noises are iteratively optimized using gradients from multiple reward models. Negative prompts generated by an LLM further calibrate the reward signal for finer compositional alignment.

perturbations $\varepsilon_t \sim \mathcal{N}(0, \mathbf{I})$ throughout the denoising trajectory, making the generation process explicitly stochastic. Accordingly, we extend the optimization objective in Eq. 1 to jointly optimize the initial latent noise $\mathbf{z}_T$ together with the intermediate perturbations $\{\varepsilon_t\}_{t=1}^T$:

$$(\mathbf{z}_T^*, \{\varepsilon_t^*\}) = \arg \max_{\mathbf{z}_T, \{\varepsilon_t\}} \mathcal{R}(\mathcal{G}_\theta(\mathbf{z}_T, \mathrm{c}; \{\varepsilon_t\})). \tag{2}$$

This full noise optimization increases the controllable degrees of freedom in the denoising process, consistently achieving higher rewards and better quality than optimizing the initial noise alone, as confirmed by our subsequent experiments.

## 3.2 REWARD MODELS FOR VIDEO GENERATION

While reward models based on human preference have been extensively explored in image generation (Kirstain et al., 2023; Xu et al., 2023; Wu et al., 2023a), their counterparts for video generation remain relatively scarce. Recent efforts such as VideoScore (He et al., 2024) and VisionReward (Xu et al., 2024) have leveraged multi-modal large language models (MLLMs) to assess the quality of generated videos. However, the sheer scale of MLLMs poses a significant barrier for direct noise optimization, particularly when gradient backpropagation through the entire generation pipeline is required. To this end, we rely on lightweight, differentiable image-level reward models and augment them with pre-trained vision-language models (VLMs) that are capable of capturing temporal dynamics. This design achieves a good balance between efficiency and effectiveness, as validated in our experiments. Specifically, we consider three types of reward objectives:

**Image-Text Alignment Reward.** We employ HPSv2 (Wu et al., 2023a) and ImageReward (Xu et al., 2023), both trained to model human preferences for image-text relevance. HPSv2 uses a CLIP-based (Radford et al., 2021) architecture with a fine-tuned reward head, while ImageReward adopts BLIP (Li et al., 2022) as its backbone, fine-tuned with supervised human feedback to improve perceptual alignment. Both models produce scalar scores that reflect the semantic alignment between a generated frame and the prompt. In practice, we compute the reward by evaluating each frame-text pair and take the mean score across all frames as the final image-text alignment reward.

**Video-Text Alignment Reward.** Image-level reward models provide strong semantic cues but lack temporal perception, making them ineffective at capturing motion consistency in video generation. To address this, we incorporate ViCLIP (Wang et al., 2023c), a video-text model trained on the large-scale InternVid dataset. ViCLIP jointly encodes short video clips and text descriptions to assess temporal and semantic consistency. In our setup, we sample two interleaved clips from the generated video, each consisting of 8 frames, and compute their similarity with the text prompt. The final video-text alignment score is obtained by averaging the two similarity scores.

**Motion Reward.** Although VLM-based rewards improve semantic alignment, they often bias models toward static content, leading to videos with limited motion and expressiveness. To mitigate this, we introduce a motion reward that explicitly accounts for temporal dynamics and motion smoothness. Specifically, we employ RAFT (Teed & Deng, 2020) to extract optical flow $\{\mathbf{F}_t\}_{t=1}^{T-1}$, where each $\mathbf{F}_t$ captures the motion from frame $t$ to $t+1$. The motion reward is then calculated as:

$$\mathcal{R}_{\text{mo}} = \tanh\Big(\underbrace{\frac{1}{T-1}\sum_{t=1}^{T-1}\|\mathbf{F}_t - \bar{\mathbf{F}}_t\|}_{\text{dynamics}}\Big) + \lambda\Big(1 - \tanh\big(\underbrace{\frac{1}{T-2}\sum_{t=1}^{T-2}\|\mathbf{F}_{t+1} - \mathbf{F}_t\|}_{\text{smoothness}}\big)\Big). \tag{3}$$

where we subtract the average flow $\bar{\mathbf{F}}_t$ in the first term to discount dominant global movement and emphasize localized dynamics The second term directly regularizes temporal changes between consecutive flows to encourage smooth and coherent motion. This objective effectively mitigates the static bias induced by image-level rewards, leading to more expressive video generation.

### 3.3 NEGATIVE-AWARE REWARD CALIBRATION

Most existing reward models (Kirstain et al., 2023; Wu et al., 2023b;a; Xu et al., 2023) are trained on human preference data using pairwise ranking, enabling them to assess which generation is "better" but not why it is better. Consequently, these models often provide limited interpretability and struggle to deliver fine-grained feedback. Prior studies (Yuksekgonul et al., 2022; Doveh et al., 2023; Trager et al., 2023) have also shown that existing VLMs behave like bag-of-words classifiers, lacking compositional reasoning capabilities. To address this limitation, we introduce a *negative-aware reward calibration* strategy. The key idea is to not only maximize alignment with the original text prompt but also explicitly enforce separation from semantically perturbed distractors.

Given a prompt $c^+$, we employ large language models (LLMs) to generate a small set of semantically perturbed hard negatives $\{c_j^-\}_{j=1}^N$, as illustrated in Figure 2. Please refer to Appendix A.4 for more details. These negatives are designed to be semantically coherent and contextually challenging, thereby encouraging the model to move beyond superficial lexical cues and instead focus on deeper semantic distinctions in video-text alignment. Formally, let $x$ denote the generated video and $\mathcal{R}(\cdot)$ be the reward model for a text-video pair. The negative-aware reward is defined as:

$$\tilde{\mathcal{R}} = \mathcal{R}(x, c^+) - \tau \cdot \log\left(\sum_{j=1}^N \exp\left(\frac{\mathcal{R}(x, c_j^-) - \mathcal{R}(x, c^+) + m}{\tau}\right)\right), \tag{4}$$

where $m$ is a margin hyperparameter and $\tau$ is a temperature coefficient. This objective guides the model to align closely with the target prompt while explicitly separating it from semantically similar distractors, thereby enhancing compositional fidelity. The effectiveness of this calibration strategy is validated by both quantitative and qualitative results.

Finally, we jointly optimize the initial noise $\mathbf{z}_T$ and the injected noise variables $\{\varepsilon_t\}_{t=1}^T$ through gradient ascent on a weighted combination of the calibrated reward objectives. For simplicity, we denote all optimized noises as $\boldsymbol{\epsilon} = \{\mathbf{z}_T, \varepsilon_1, \ldots, \varepsilon_T\}$, and the overall update rule is given by:

$$\boldsymbol{\epsilon}^{k+1} = \boldsymbol{\epsilon}^k + \eta \cdot \nabla_{\boldsymbol{\epsilon}}\left[\sum_i \alpha_i \tilde{\mathcal{R}}_i(\mathcal{G}_\theta(\boldsymbol{\epsilon}, \mathbf{c}), \mathbf{c})\right], \tag{5}$$

where $k$ is the optimization step, $\eta$ is the learning rate, and $\alpha_i$ are weights for individual reward components. In practice, we find that 25 steps already yield substantial improvements, striking a good balance between performance and computational cost while avoiding reward over-optimization. By combining gradient checkpointing and mixed-precision training, we make it feasible to run noise optimization on just two RTX 6000 Ada GPUs in about 45 seconds per sample, further reduced to 30 seconds on a single H100 GPU. Under the same time budget, our method even surpasses large-scale pretrained models such as CogVideoX-2B (Yang et al., 2024b) and Wan2.1-1.3B (Wan et al., 2025) in human preference rate, further highlighting its efficiency and effectiveness.

Table 1: Quantitative Evaluation on **VBench** (%). * denotes our reproduction results. The best results among open-source models are highlighted in **bold** and the second best is underlined.

| Model | Avg. Score | Overall Consist. | Multiple Objects | Color | Spatial Relation. | Motion Smooth. | Dynamic Degree |
|---|---|---|---|---|---|---|---|
| *(close-source)* | | | | | | | |
| Pika 1.0 | 61.27 | 25.94 | 43.08 | 90.57 | 61.03 | 99.50 | 47.50 |
| Gen-3 | 64.28 | 26.69 | 53.64 | 80.90 | 65.09 | 99.23 | 60.14 |
| Kling | 69.67 | 26.42 | 68.05 | 89.90 | 73.03 | 99.40 | 61.21 |
| Sora | 71.69 | 26.26 | 70.85 | 80.11 | 74.29 | 98.74 | 79.91 |
| Veo 3 | 74.74 | 27.88 | 82.20 | 82.48 | 84.26 | 99.16 | 72.43 |
| *(open-source)* | | | | | | | |
| ModelScope | 57.04 | 25.67 | 38.98 | 81.72 | 33.68 | 95.79 | 66.39 |
| VideoCrafter2 | 56.32 | 28.23 | 40.66 | 92.92 | 35.86 | 97.73 | 42.50 |
| CogVideoX-2B | 68.92 | 27.33 | 66.59 | 83.01 | 74.27 | 97.51 | 64.79 |
| Open-Sora 2.0 | **72.91** | 27.50 | 77.72 | 85.98 | **76.18** | 98.69 | **71.39** |
| Vchitect-2.0 | 67.31 | 27.57 | 68.84 | 87.04 | 57.55 | **98.98** | 63.89 |
| T2V-Turbo (MS) | 63.93 | 27.51 | 58.63 | 89.67 | 45.74 | 95.64 | 66.39 |
| T2V-Turbo (MS)* | 64.13 | 27.40 | 54.50 | 89.90 | 46.62 | 95.51 | 70.83 |
| **+ NoisEasier (Ours)** | 69.78 | **32.05** | 76.62 | **94.38** | 48.93 | 95.29 | **71.39** |
| T2V-Turbo (VC2) | 59.65 | 28.16 | 54.65 | 89.90 | 38.67 | 97.34 | 49.17 |
| T2V-Turbo (VC2)* | 60.16 | 28.12 | 54.33 | 90.59 | 39.95 | 96.29 | 51.67 |
| **+ NoisEasier (Ours)** | 68.77 | 31.95 | **79.18** | 93.63 | 52.61 | 96.10 | 59.17 |

## 4 EXPERIMENTS

**Experimental Setup.** We implement NoisEasier on two video consistency backbones: T2V-Turbo (MS) and T2V-Turbo (VC2), distilled respectively from ModelScope (Wang et al., 2023a) and VideoCrafter2 (Chen et al., 2024a). These lightweight variants support efficient sampling within 4 to 8 denoising steps; considering the computational budget, we use 4 steps for fast generation. Noise optimization is performed over 25 steps using AdamW with gradient clipping and a learning rate of 0.01. We evaluate NoisEasier on two widely used benchmarks: VBench (Huang et al., 2024) and T2V-CompBench (Sun et al., 2024). VBench offers a broad evaluation across 16 video quality dimensions. We report results on four compositional dimensions (overall consistency, multiple objects, color, and spatial relation) and two motion dimensions (motion smoothness and dynamic degree). T2V-CompBench is tailored for compositional generation, assessing seven categories such as consistent attribute binding, motion binding, and spatial relationships. More implementation details can be found in Appendix A.5.

### 4.1 QUANTITATIVE EVALUATION

We report quantitative results on VBench in Table 1, comparing with both closed-source models (i.e., Pika (Pika Team, 2024), Gen-3 (Runway, 2024), Kling (Kuaishou, 2024), Sora (Brooks et al., 2024), and Veo 3 (Google, 2025)) and open-source baselines (ModelScope (Wang et al., 2023a), VideoCrafter2 (Chen et al., 2024a), CogVideoX-2B (Yang et al., 2024b), Open-Sora 2.0 (Zheng et al., 2024), and Vchitect-2.0 (Fan et al., 2025)). Notably, NoisEasier consistently improves T2V-Turbo baselines by great margins, with average gains of +5.65 (MS) and +8.61 (VC2), even outperforming the closed-source model Kling. The largest improvements are observed in Multiple Objects, followed by Spatial Relation and Consistency, while Color shows moderate gains, highlighting the model's strength in multi-entity reasoning and relational compositionality. Although Motion Smoothness drops slightly, the substantial gain in Dynamic Degree suggests that NoisEasier achieves a favorable trade-off, producing richer motion while largely preserving temporal coherence. On T2V-CompBench, NoisEasier also brings significant gains. As shown in Table 2, Action Binding improves from 0.49 to 0.65 (MS) and 0.62 to 0.75 (VC2), with clear improvements in Consistency Attribute, Interaction, Spatial, and Numeracy. These results further demonstrate that our framework enhances both static compositional fidelity and temporal relational coherence.

Table 2: Quantitative Evaluation on **T2V-CompBench**. The best results among open-source models are highlighted in **bold** and the second best is underlined.

| Model | Consist Attr. | Dynamic Attr. | Spatial | Motion | Action | Interaction | Numeracy |
|---|---|---|---|---|---|---|---|
| *(close-source)* | | | | | | | |
| Pika 1.0 | 0.5536 | 0.0128 | 0.4650 | 0.2234 | 0.4250 | 0.5198 | 0.3870 |
| Gen-3 | 0.5980 | 0.0687 | 0.5194 | 0.2754 | 0.5233 | 0.5906 | 0.2306 |
| Dreamina | 0.6913 | 0.0051 | 0.5773 | 0.2361 | 0.5924 | 0.6824 | 0.4380 |
| PixVerse | 0.7060 | 0.0624 | 0.5979 | 0.2867 | 0.8722 | 0.8309 | 0.6066 |
| Kling | 0.6931 | 0.0098 | 0.5690 | 0.2562 | 0.5787 | 0.7128 | 0.4413 |
| *(open-source)* | | | | | | | |
| ModelScope | 0.5148 | 0.0161 | 0.4118 | 0.2408 | 0.3639 | 0.4613 | 0.1986 |
| Show-1 | 0.5670 | 0.0115 | 0.4544 | 0.2291 | 0.3881 | 0.6244 | 0.3086 |
| VideoTetris | 0.6211 | 0.0104 | 0.4832 | 0.2249 | 0.4939 | 0.6578 | 0.3467 |
| VideoCrafter2 | 0.6182 | 0.0103 | 0.4838 | 0.2259 | 0.5030 | 0.6365 | 0.3330 |
| Open-Sora 1.2 | 0.5639 | 0.0189 | 0.5063 | 0.2468 | 0.4833 | 0.5039 | 0.3719 |
| CogVideoX-5B | 0.6164 | **0.0219** | 0.5172 | **0.2658** | 0.5333 | 0.6069 | 0.3706 |
| T2V-Turbo (MS)* | 0.7313 | 0.0124 | 0.4620 | 0.2306 | 0.4923 | 0.5822 | 0.3030 |
| **+ NoisEasier (Ours)** | 0.8375 | 0.0147 | 0.5259 | 0.2502 | 0.6477 | 0.6743 | **0.3983** |
| T2V-Turbo (VC2)* | 0.7852 | 0.0113 | 0.5079 | 0.2222 | 0.6241 | 0.7017 | 0.2900 |
| **+ NoisEasier (Ours)** | **0.8737** | 0.0097 | **0.5660** | 0.2343 | **0.7524** | **0.7915** | 0.3753 |

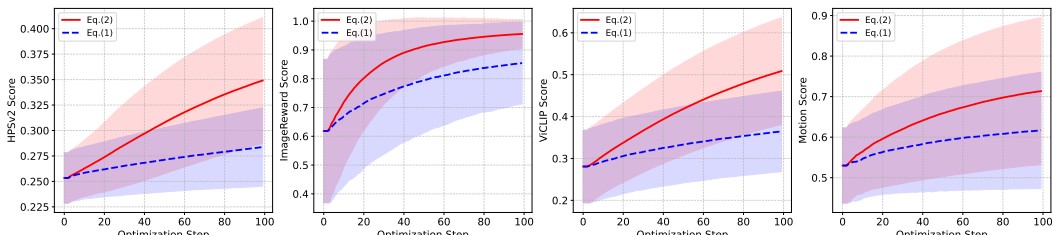

Figure 3: Reward curve during noise optimization. The red solid line represents full noise optimization, while the blue dotted line denotes initial noise optimization. All scores are scale-normalized.

## 4.2 ABLATION STUDY

**Effect of Various Reward Models.** As shown in Table 3, we observe that individual reward models specialize in different aspects: among all reward models, ImageReward and ViCLIP contribute most to semantic alignment. However, optimizing only semantic rewards often yields overly static videos, as evidenced by the decreased Dynamic Degree score. In contrast, the motion reward alone boosts both temporal dynamics and coherence, yet degrades all other metrics, underscoring the necessity of synergies across reward objec-

Table 3: The effect of reward models for NoisEasier on VBench. The best score among individual reward models is marked in green, while the second-best is in yellow.

| Model | Overall Consist. | Multiple Objects | Color | Spatial Relation. | Motion Smooth. | Dynamic Degree |
|---|---|---|---|---|---|---|
| T2V-Turbo (MS)* | 27.40 | 54.50 | 89.90 | 46.62 | 95.51 | 70.83 |
| + HPSv2 | 27.85 | 64.66 | 90.20 | 46.21 | 95.19 | 54.72 |
| + ImageReward | 28.24 | 73.78 | 90.27 | 47.16 | 95.36 | 61.11 |
| + ViCLIP | 36.21 | 68.05 | 91.19 | 41.92 | 94.98 | 63.89 |
| + Motion | 26.05 | 47.68 | 87.32 | 42.07 | 95.59 | 85.56 |
| All w/o negatives | 31.74 | 75.82 | 93.29 | 47.95 | 95.38 | 73.33 |
| All (NoisEasier) | 32.05 | 76.62 | 94.38 | 48.93 | 95.29 | 71.39 |

tives. We further provide a qualitative analysis of the dynamic term in the motion reward, as illustrated in Figure 4a. By subtracting the average flow in Eq. 3, the residual motion effectively suppresses camera shifts while emphasizing subject movement. When combining all rewards, we observe complementary gains across dimensions. The negative-aware reward calibration further enhances semantic alignment, though motion-related scores slightly decrease, as the calibration mainly targets VLM-based objectives. More combination ablations can be found in Appendix A.2.

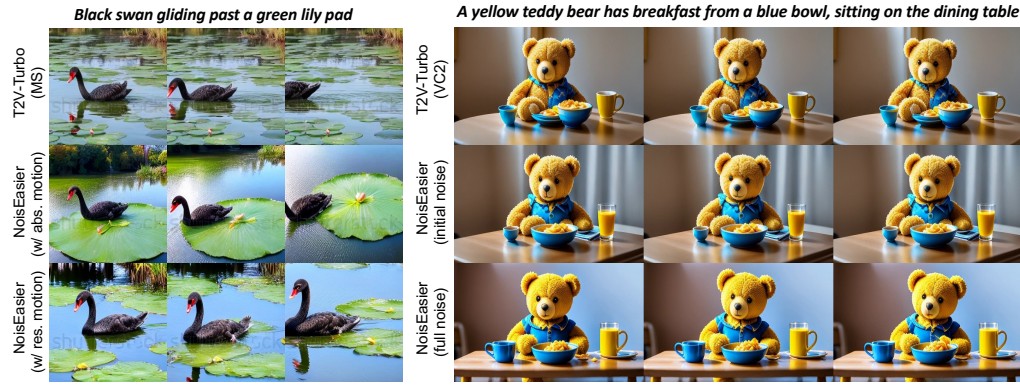

(a) Absolute motion *vs.* Residual motion.  (b) Initial noise optimization *vs.* Full noise optimization.

Figure 4: Case study of the effect of (a) motion reward; and (b) noise optimization formulation.

**Initial *vs.* Full noise optimization.** To evaluate the advantage of full noise optimization, we compare with its counterpart where only the initial latent noise is updated. As shown in Table 4, full noise optimization significantly outperforms initial-only optimization across all VBench dimensions.

Table 4: Effect of different noise optimization options.

| Method | Overall Consist. | Multiple Objects | Color | Spatial Relation. | Motion Smooth. | Dynamic Degree |
|---|---|---|---|---|---|---|
| Initial noise | 29.67 | 72.58 | 90.03 | 44.10 | 95.58 | 66.94 |
| Full noise | 32.05 | 76.62 | 94.38 | 48.93 | 95.29 | 71.39 |

Figure 3 further illustrates this effect: across all reward models, the red curves consistently lie above the blue ones, and the gap widens as optimization progresses. This indicates that optimizing intermediate noises provides additional flexibility for aligning the denoising trajectory with reward signals. We also provide an example in Figure 4b to illustrate the difference: the initial noise primarily governs global appearance and coarse layout, while the intermediate noises shape finer structures and stylistic details, leading to improved aesthetics with sharper contrast and more visually appealing results. These results confirm the effectiveness of full noise optimization in enhancing both quantitative performance and visual quality.

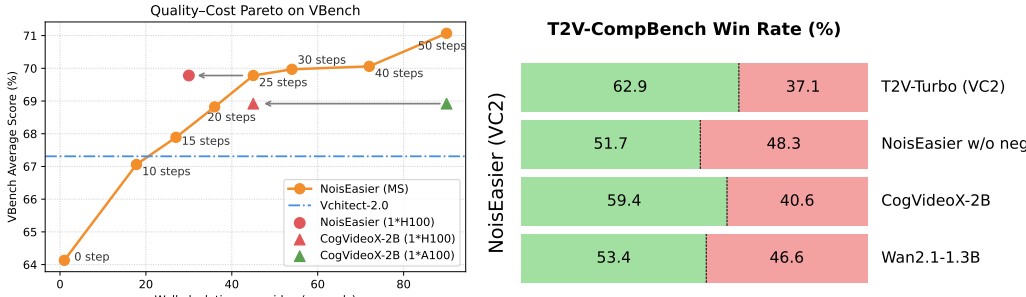

Figure 5: Quality-Cost Pareto on VBench.  Figure 6: User study results on T2V-CompBench.

**Performance and Efficiency.** Figure 5 presents the quality-cost trade-off between inference time and VBench score. While our method introduces additional inference time compared to vanilla sampling, it consistently pushes the Pareto frontier forward. For example, with 25 steps, T2V-Turbo (MS) with NoisEasier achieves an average score of 69.78 on VBench, outperforming the competitive CogVideoX-2B while requiring less or comparable wall-clock time. While optimizing for 50 steps can further increase the score to 71.07, it also doubles the wall-clock time, making it less efficient and less favorable for real-world usage. Therefore, we adopt 25 steps as the main operating point. Notably, T2V-Turbo already utilized HPSv2 and ViCLIP for reward-based model distillation, yet our results show that test-time optimization can still yield substantial improvements, indicating that offline fine-tuning does not fully capture the reward signal. This highlights the effectiveness of test-time optimization as a lightweight and complementary alternative to offline approaches.

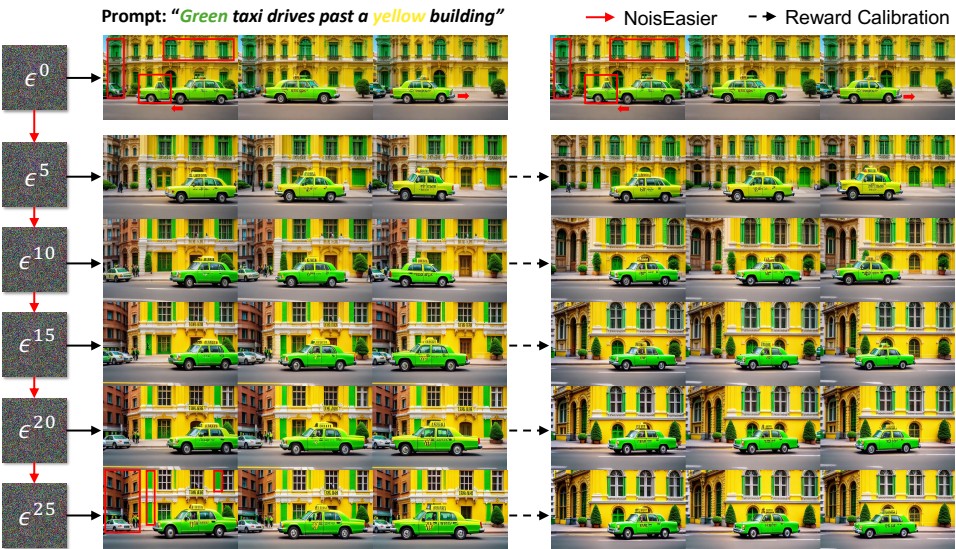

Figure 7: Sample visualization of intermediate steps during noise optimization.

### 4.3 HUMAN EVALUATION

We further conduct a user study to evaluate the advantages of our method. Specifically, we sample 70 test prompts from T2V-CompBench and collect results from four competing methods. Each sample is evaluated by five Amazon Mechanical Turk workers, who are asked to select the candidate that achieves better text-to-video alignment with good visual quality. As shown in Figure 6, NoisEasier achieves a notable win rate of 62.9% against the baseline, indicating that noise optimization significantly improves compositional alignment. Incorporating negative prompts yields further improvements in human preference, albeit with a modest margin. Moreover, against recent large-scale pretrained models such as CogVideoX-2B and Wan2.1-1.3B, NoisEasier still maintains a competitive edge. More results and evaluation details can be found in Appendix A.3.

### 4.4 QUALITATIVE ANALYSIS

Finally, we provide qualitative comparisons in Figure 7 to illustrate the effect of our optimization framework. The baseline captures the rough semantics of the prompt but suffers from clear compositional flaws, including incorrect color binding, inconsistent object and motion. Applying noise optimization gradually improves both spatial and temporal quality, i.e., the taxi becomes a coherent object with stable right-to-left motion, and attribute separation improves, though some artifacts like green stripes on the facade persist. Negative-aware reward calibration further enhances semantic disentanglement: the facade is rendered fully yellow, including the miscolored left-side regions, and the green taxi is more distinctly preserved with minimal leakage. More visualization results can be found in Figure 1 and Appendix A.6.

## 5 CONCLUSION

In this work, we propose NoisEasier, a test-time training framework that improves text-to-video generation by directly optimizing latent noise with differentiable rewards. Without any model fine-tuning, it achieves substantial compositional gains while preserving motion quality, outperforming both commercial and open-source baselines within a one-minute inference budget. Our results show that optimizing the full noise trajectory improves human preference, while combining semantic and motion rewards prevents static overfitting. The proposed negative-aware reward calibration further enhances fine-grained alignment through LLM-generated distractors. Overall, this work highlights the potential of test-time noise optimization as a lightweight yet effective alternative to reward-guided fine-tuning, and we hope it encourages further exploration of test-time controllability in generative video models.

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

# A  APPENDIX

## A.1  PRELIMINARIES

**Diffusion Models (DMs)** (Ho et al., 2020; Song et al., 2020b; Karras et al., 2022) have emerged as a powerful class of generative models that synthesize high-quality samples by gradually transforming data into noise and learning to reverse the process. Formally, the forward process can be described by a stochastic differential equation (SDE) that perturbs a clean sample $\mathbf{x}_0 \sim p_0(\mathbf{x})$ into Gaussian noise over a time horizon $t \in [0, T]$:

$$\mathrm{d}\mathbf{x}_t = f(\mathbf{x}_t, t)\,\mathrm{d}t + g(t)\,\mathrm{d}\mathbf{w}_t, \tag{6}$$

where $\mathbf{w}_t$ denotes standard Wiener process, and $f(\cdot, t)$ and $g(t)$ control the drift and diffusion strength, respectively. With appropriate schedules, the final state $\mathbf{x}_T$ approaches the isotropic Gaussian distribution $\mathcal{N}(0, \sigma^2(T)\mathbf{I})$. To generate samples, one must reverse this stochastic trajectory from noise back to the data manifold, leading to a reverse-time SDE (Anderson, 1982):

$$\mathrm{d}\mathbf{x}_t = \left[f(\mathbf{x}_t, t) - g(t)^2\,\nabla_{\mathbf{x}} \log p_t(\mathbf{x}_t)\right]\mathrm{d}t + g(t)\,\mathrm{d}\bar{\mathbf{w}}_t, \tag{7}$$

where $\nabla_{\mathbf{x}} \log p_t(\mathbf{x}_t)$ is the score function and $\bar{\mathbf{w}}_t$ is a reverse-time Wiener process. Since the exact score is intractable, a neural network $\epsilon_\theta(\mathbf{x}_t, t)$ is trained to approximate it via denoising score matching. This leads to a practical sampling process, which can also be expressed as an ordinary differential equation by removing the stochastic term, known as Probability Flow (PF-ODE):

$$\mathrm{d}\mathbf{x}_t = \left[f(\mathbf{x}_t, t) - \frac{1}{2}g(t)^2\,\epsilon_\theta(\mathbf{x}_t, t)\right]\mathrm{d}t. \tag{8}$$

This ODE formulation enables deterministic sampling using numerical solvers (Lu et al., 2022), offering compatibility with adaptive step sizes and efficient inference. In practice, the model is trained using a mean squared error loss derived from the closed-form solution of the forward process. Assuming $\mathbf{x}_t = \alpha_t \mathbf{x}_0 + \sigma_t \boldsymbol{\varepsilon}$, with $\boldsymbol{\varepsilon} \sim \mathcal{N}(0, \mathbf{I})$, the training objective is denoted as:

$$\mathcal{L}_{\mathrm{DM}}(\theta) = \mathbb{E}_{t, \mathbf{x}_0, \boldsymbol{\varepsilon}}\left[\|\boldsymbol{\varepsilon} - \epsilon_\theta(\mathbf{x}_t, t)\|^2\right]. \tag{9}$$

This framework underpins many state-of-the-art image and video generation models. In T2V generation, the diffusion typically operates in a latent space, where each sample $\mathbf{z}_t \in \mathbb{R}^{C \times T \times H \times W}$ is a spatiotemporal tensor. The denoising network $\epsilon_\theta(\mathbf{z}_t, \mathbf{c}, t)$ is conditioned on a text prompt $\mathbf{c}$ and implemented using a 3D U-Net (Blattmann et al., 2023; Guo et al., 2023) or Transformer (Lu et al., 2023; Ma et al., 2024; Yang et al., 2024b) backbone. During inference, samples are generated by discretizing and solving the SDE or ODE over a finite number of time steps.

**Consistency Models.** While diffusion models achieve strong generation performance, their sampling process typically involves solving the reverse-time SDE or ODE over hundreds of steps, resulting in high inference cost. To alleviate this, *consistency models* (CMs) (Song et al., 2023; Luo et al., 2023) have been proposed to replace iterative sampling with a one-step mapping from noised inputs to clean outputs. Instead of simulating the full trajectory of a diffusion process, a CM learns a function that maps an intermediate state $\mathbf{x}_t$ to its corresponding origin $\mathbf{x}_\kappa$ on the PF-ODE trajectory: $f : (\mathbf{x}_t, t) \mapsto \mathbf{x}_\kappa, t \in [\kappa, T]$, where $\kappa$ is a small positive constant. Accordingly, the model is trained to enforce *self-consistency*, which means the output should be invariant across different time inputs from the same PF-ODE trajectory:

$$f(\mathbf{x}_t, t) = f(\mathbf{x}_{t'}, t'), \quad \forall t, t' \in [\kappa, T], \tag{10}$$

where $\mathbf{x}_t$ and $\mathbf{x}_{t'}$ are two noised states originating from the same ground truth sample. This eliminates the need for sequential denoising steps at inference and allows for fast, one-step generation.

Consistency models are typically parameterized with skip connections to ensure smooth interpolation between input and output across time. A common architecture adopts a time-dependent interpolation:

$$f_\theta(\mathbf{x}, t) = c_{\mathrm{skip}}(t)\,\mathbf{x} + c_{\mathrm{out}}(t)\,F_\theta(\mathbf{x}, t), \tag{11}$$

where $F_\theta$ is a neural network, and $c_{\mathrm{skip}}(t), c_{\mathrm{out}}(t)$ are time-dependent weighting functions satisfying $c_{\mathrm{skip}}(\kappa) = 1, c_{\mathrm{out}}(\kappa) = 0$ to enforce the boundary condition $f(\mathbf{x}_\kappa, \kappa) = \mathbf{x}_\kappa$.

## A.2 MORE ABLATIONS FOR REWARD MODELS

In this section, we present all the combinations of the reward models in Table 5. Among single-reward settings, ViCLIP delivers the strongest gains in overall consistency and color fidelity, highlighting its strength in semantic alignment and global appearance, whereas ImageReward performs best on multiple objects and spatial relations, reflecting its training on fine-grained visual-text cues. The motion reward is more distinctive: although it trails others on appearance-related metrics, it substantially boosts dynamic degree and smoothness, underscoring its importance in preventing static generations. When combining rewards, synergies are clear: pairing the motion reward with ViCLIP or ImageReward strikes a strong balance between temporal dynamics and spatial fidelity, while conflicts also appear; for instance, HPSv2 with other rewards can suppress motion expressiveness even as it improves color quality. Notably, the best three-reward combinations (e.g., ImageReward + ViCLIP + Motion) nearly match the four-reward setting. Overall, the results show that different reward models specialize in complementary dimensions, and that careful combinations are crucial to balancing fidelity, compositionality, and temporal quality.

Table 5: All Combinations of reward models applied to NoisEasier (MS) on VBench. The best score for each number of reward models is highlighted in green, and the second-best in yellow.

| HPSv2 | ImgReward | ViCLIP | Motion | Overall Consist. | Multiple Objects | Color | Spatial Relation. | Motion Smooth. | Dynamic Degree |
|---|---|---|---|---|---|---|---|---|---|
| ✗ | ✗ | ✗ | ✗ | 27.40 | 54.50 | 89.90 | 46.62 | 95.51 | 70.83 |
| ✓ | ✗ | ✗ | ✗ | 27.85 | 64.66 | 90.20 | 46.21 | 95.19 | 54.72 |
| ✗ | ✓ | ✗ | ✗ | 28.24 | 73.78 | 90.27 | 47.16 | 95.36 | 61.11 |
| ✗ | ✗ | ✓ | ✗ | 36.21 | 68.05 | 91.19 | 41.92 | 94.98 | 63.89 |
| ✗ | ✗ | ✗ | ✓ | 26.05 | 47.68 | 87.32 | 42.07 | 95.59 | 85.56 |
| ✓ | ✓ | ✗ | ✗ | 28.40 | 74.02 | 89.35 | 47.18 | 95.53 | 55.28 |
| ✓ | ✗ | ✓ | ✗ | 34.57 | 70.78 | 92.63 | 43.87 | 94.77 | 58.89 |
| ✓ | ✗ | ✗ | ✓ | 27.28 | 58.28 | 89.31 | 44.09 | 95.52 | 76.67 |
| ✗ | ✓ | ✓ | ✗ | 32.31 | 71.75 | 90.69 | 46.89 | 95.31 | 62.78 |
| ✗ | ✓ | ✗ | ✓ | 28.03 | 73.26 | 89.34 | 44.43 | 95.43 | 74.72 |
| ✗ | ✗ | ✓ | ✓ | 34.61 | 64.04 | 89.28 | 43.53 | 94.95 | 80.28 |
| ✓ | ✓ | ✓ | ✗ | 31.95 | 74.09 | 92.01 | 46.78 | 95.30 | 59.44 |
| ✓ | ✓ | ✗ | ✓ | 28.20 | 73.16 | 91.50 | 46.36 | 95.38 | 75.28 |
| ✓ | ✗ | ✓ | ✓ | 33.46 | 71.39 | 91.54 | 47.26 | 95.19 | 76.11 |
| ✗ | ✓ | ✓ | ✓ | 31.96 | 74.82 | 92.22 | 47.49 | 95.35 | 75.56 |
| ✓ | ✓ | ✓ | ✓ | 31.74 | 75.82 | 93.29 | 47.95 | 95.38 | 73.33 |

## A.3 HUMAN EVALUATION

We conducted a human evaluation study on Amazon Mechanical Turk (MTurk) to compare pairs of AI-generated videos. Specifically, we uniformly sample 10 prompts from each of the 7 dimensions of T2V-CompBench, yielding 70 test prompts in total. We compare NoisEasier with four models: the baseline T2V-Turbo (VC2), NoisEasier (VC2) without reward calibration, CogVideoX-2B, and Wan2.1-1.3B. To ensure fairness, we use the same random seed for each prompt across models. For each trial, workers were shown a text description along with two candidate videos (Video A and Video B) generated from the same prompt, as illustrated in Figure 9. The order of the two videos was randomly shuffled to avoid positional bias, and each trial was independently evaluated by five workers. To help workers better understand the task, we provided the following instructions:

- **Text-to-Video Alignment:** Does the video correctly depict the objects, actions, attributes, relations, and numeracy in the prompt? If both are inaccurate, choose the one with fewer misalignments.

- **Visual Quality:** Which video shows fewer artifacts, distortions, or temporal glitches?

- **Overall Preference:** Considering alignment and visual quality together, which video is preferred overall?

In addition to the overall preference reported in Figure 6, we present comparison results on semantic alignment and visual quality in Figure 8. Workers consistently preferred NoisEasier over T2V-Turbo in both dimensions, confirming the effectiveness of noise optimization beyond baseline tuning. Notably, when comparing NoisEasier with and without negative prompts, visual quality remains nearly identical, but semantic alignment preference improves by 5.2%, demonstrating the clear benefit of reward calibration. Against large-scale pretrained models such as CogVideoX-2B and Wan2.1-1.3B, NoisEasier remains competitive, with around 60% alignment preference and 52-55% visual quality preference, despite being a lighter optimization strategy. Overall, these results suggest that reward-calibrated noise optimization not only enhances semantic alignment but also offers a favorable trade-off with visual quality compared to both baselines and larger-scale models.

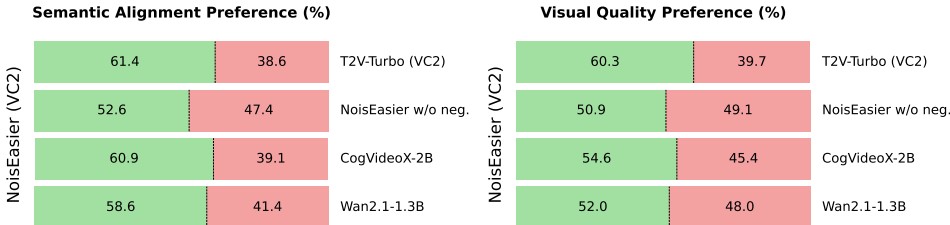

(a) User preference for T2V alignment.          (b) User preference for visual quality.

Figure 8: User study results on text-to-video alignment and visual quality on T2V-CompBench.

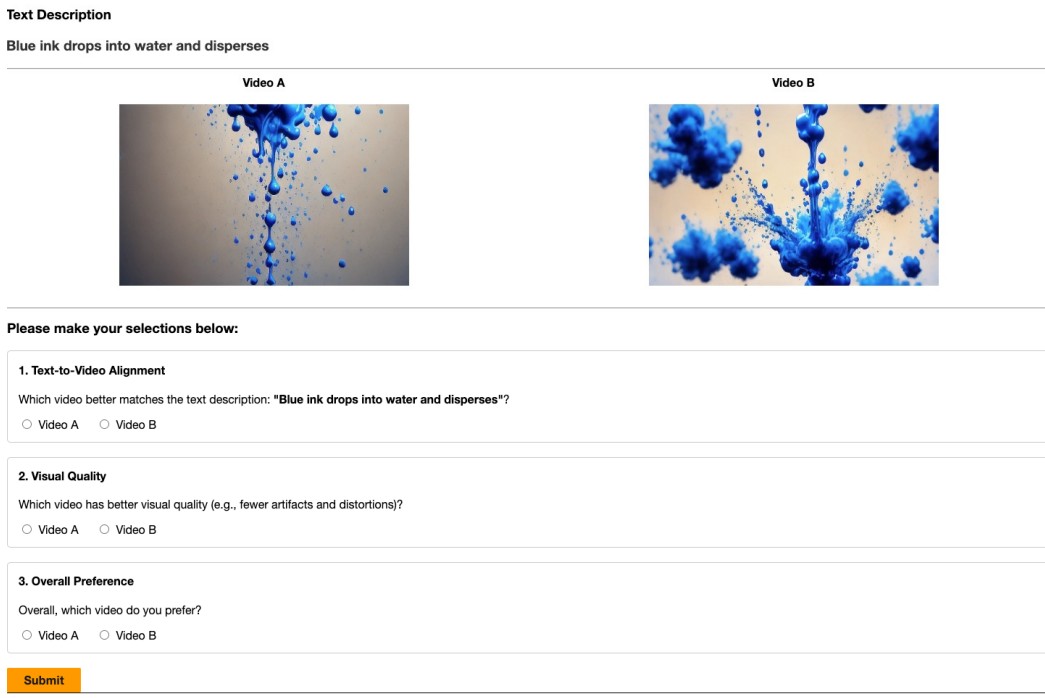

Figure 9: User Evaluation Interface in Amazon Mechanical Turk.

## A.4 INSTRUCTION OF NEGATIVE PROMPT GENERATION VIA LLM

To construct hard negatives for reward calibration, we designed an automated pipeline using the OpenAI API. Figure 10 illustrates the prompt template along with several generated examples. Given an input prompt from a specific dimension, the pipeline iterates through each prompt and queries GPT-4o with explicit instructions to generate syntactically similar but semantically different

variants. Negatives are produced by modifying only one category at a time (e.g., subject, object, or action), ensuring that changes reflect clear semantic shifts rather than trivial rephrasings or synonyms. For each prompt, the model returns a fixed number of negatives (eight by default) in strict JSON format, which are automatically parsed, validated, and saved into a consolidated JSON file. This process yields a large pool of structured negatives that are consistent in format and directly usable for contrastive learning and evaluation.

To further validate our negative prompt construction strategy, we also experimented with randomly sampled negatives generated via part-of-speech (POS) tagging. Specifically, we applied spaCy[1] to identify core primitives (verbs, nouns, adjectives, adpositions, and numerals) and randomly replaced them with alternatives drawn from the test prompt suite. Table 6 compares this random sampling approach against our LLM-generated negatives. While both methods yield broadly similar trends, LLM-generated negatives consistently achieve slightly higher scores across most dimensions, particularly in multiple objects and spatial relations, suggesting that structured generation produces more semantically meaningful contrasts. Random sampling, though simpler, often introduces arbitrary replacements that may not capture true compositional conflicts, which limits its effectiveness. These results confirm that carefully designed hard negatives from LLMs are more effective for reward calibration than naive random substitutions.

Table 6: Effect of different negative prompt construction strategies.

| Method | Overall Consist. | Multiple Objects | Color | Spatial Relation. | Motion Smooth. | Dynamic Degree |
|---|---|---|---|---|---|---|
| Random sample | 31.83 | 75.41 | 93.14 | 46.39 | 95.46 | 71.11 |
| LLM-generated | 32.05 | 76.62 | 94.38 | 48.93 | 95.29 | 71.39 |

## A.5 IMPLEMENTATION DETAILS

Considering that different reward models produce outputs on different scales, we assign them separate weights $\alpha$ in Eq. 5. Specifically, HPSv2 and ViCLIP measure cosine similarity between text and video, with most values falling in the range of 0.2–0.4, so their weights are set to 2. The raw scores of ImageReward lie within $[-2, +2]$; we normalize them to $[0, 1]$ using min-max scaling to ensure non-negative rewards, and assign a weight of 1. For the motion reward, both the dynamic and smoothness terms are in $[0, 1]$, with $\lambda = 0.5$ and an overall weight of 1. We use GPT-4o to generate 8 negative samples per prompt. For negative-aware reward calibration, we set $m = 0.02$ and $\tau = 0.01$ for HPSv2, and $m = \tau = 0.05$ for ViCLIP. We keep the original ImageReward since it is not based on video-text contrastive pretraining, and this also helps save memory and maintain inference speed. On both benchmarks, we empirically apply the above default settings for the two baseline models, which already demonstrate strong performance. Our code will be open-source upon acceptance.

## A.6 MORE VISUALIZATION RESULTS

We provide more qualitative examples in Figure 11 and Figure 12 to illustrate the effectiveness of our method.

## A.7 LLM USAGE

We used the GPT model in two ways: (1) for grammar checking and minor language polishing of the manuscript, and (2) for generating negative prompts that were used in our reward calibration experiments, as clarified in Appendix A.4. No content generation related to ideation or methodology design was performed by LLMs.

---

[1]spaCy: https://spacy.io/

**System-Msg**: "You are an assistant for generating hard negative prompts for text-to-video generation. Given an original text prompt and a set of compositional categories, your task is to generate a specified number of hard negative prompts. Negatives should be generated by applying operations such as replacement, swapping, or modification across specified categories like subject, object, color, shape, spatial relationship, numeracy, action, and scenario. First, identify which of the provided categories are explicitly present in the original prompt. Only modify categories that are present. "

For each negative prompt:
- Modify exactly ONE present category per sample; all other elements must remain unchanged.
- Keep the syntax and structure consistent with the original prompt.
- Ensure the modification is semantically contrastive. Do not use synonyms, near-synonyms, or trivial rephrasing.
- The output must strictly follow this JSON format: "{\n
"original_prompt: \"<original_prompt>\", "negative_prompt": [\"negative1\",
\"negative2\", ..., \"negativeN\"]\n"}"

**Input:**
Original Prompt: "A blue car drives past a white picket fence on a sunny day"
Category to modify: ["color"]
Number of negative prompts: 8

**Output:**
{"original_prompt": "A blue car drives past a white picket fence on a sunny day",
"negative_prompt": ["A red car drives past a white picket fence on a sunny day", "A green car drives past a white picket fence on a sunny day", "A black car drives past a white picket fence on a sunny day", "A yellow car drives past a white picket fence on a sunny day", "A blue car drives past a black picket fence on a sunny day", "A blue car drives past a brown picket fence on a sunny day", "A blue car drives past a red picket fence on a sunny day", "A blue car drives past a green picket fence on a sunny day"]}

**Input:**
Original Prompt: "A dog running on the left of a bicycle"
Category to modify: ["spatial relationship", "object"]
Number of negative prompts: 8

**Output:**
{"original_prompt": "A dog running on the left of a bicycle", "negative_prompt": ["A dog running on the right of a bicycle", "A dog running in front of a bicycle", "A dog running behind a bicycle", "A dog running under a bicycle", "A cat running on the left of a bicycle", "A bird running on the left of a bicycle", "A dog running on the left of a car", "A dog running on the left of a rock"]}

**Input:**
Original Prompt: "One tree stands tall, swaying gently in the breeze."
Category to modify: ["numeracy"]
Number of negative prompts: 8

**Output:**
{"original_prompt": "One tree stands tall, swaying gently in the breeze.", "negative_prompt": ["Two trees stand tall, swaying gently in the breeze.", "Three trees stand tall, swaying gently in the breeze.", "Four trees stand tall, swaying gently in the breeze.", "Five trees stand tall, swaying gently in the breeze.", "Six trees stand tall, swaying gently in the breeze.", "Seven trees stand tall, swaying gently in the breeze.", "Eight trees stand tall, swaying gently in the breeze.", "Nine trees stand tall, swaying gently in the breeze."]}

Figure 10: Prompt template and generated negative samples of GPT-4o.

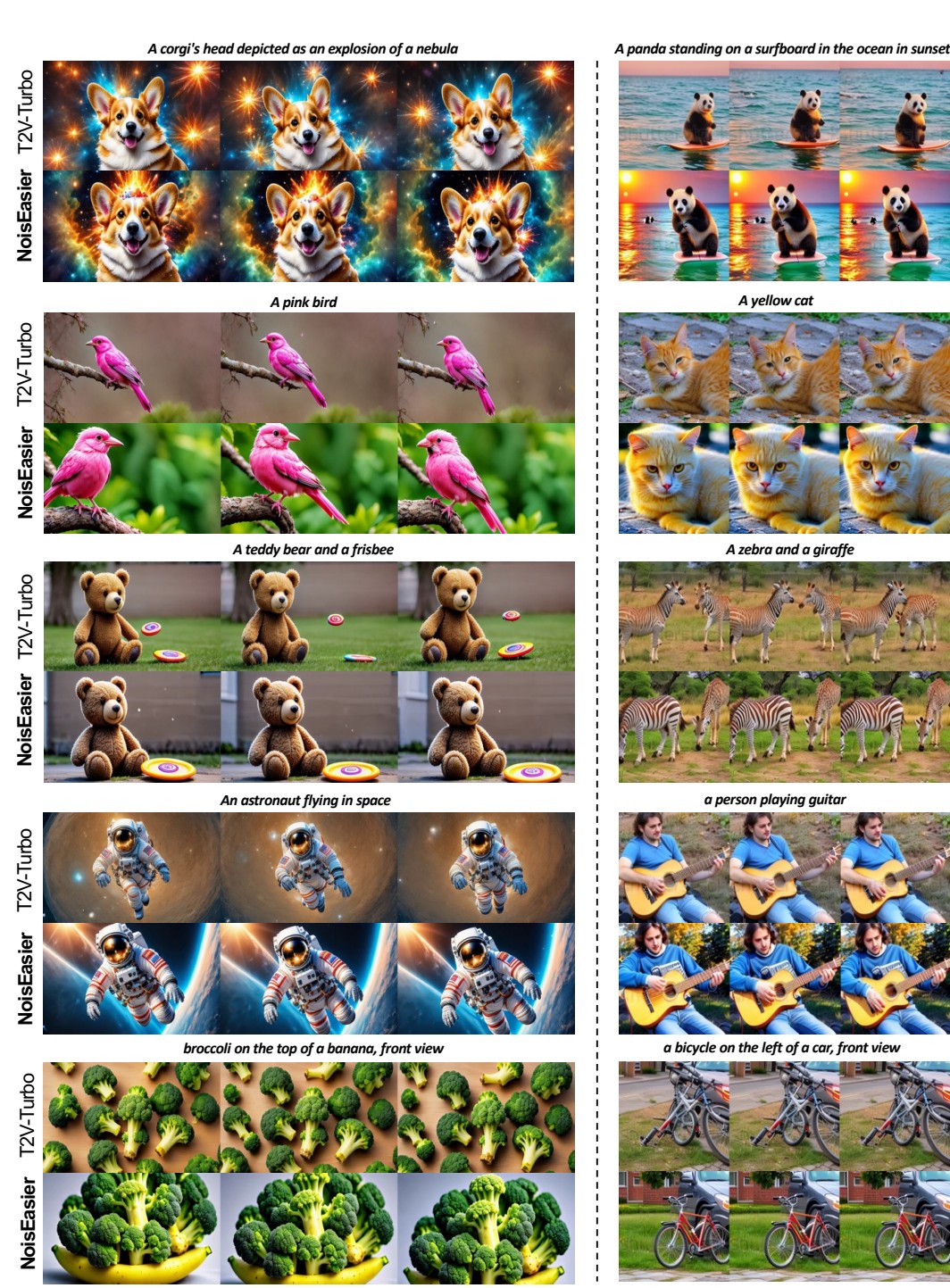

Figure 11: Qualitative results with and without NoisEasier on VBench. **Left**: T2V-Turbo (VC2), 4-step generation; **Right**: T2V-Turbo (MS), 4-step generation.

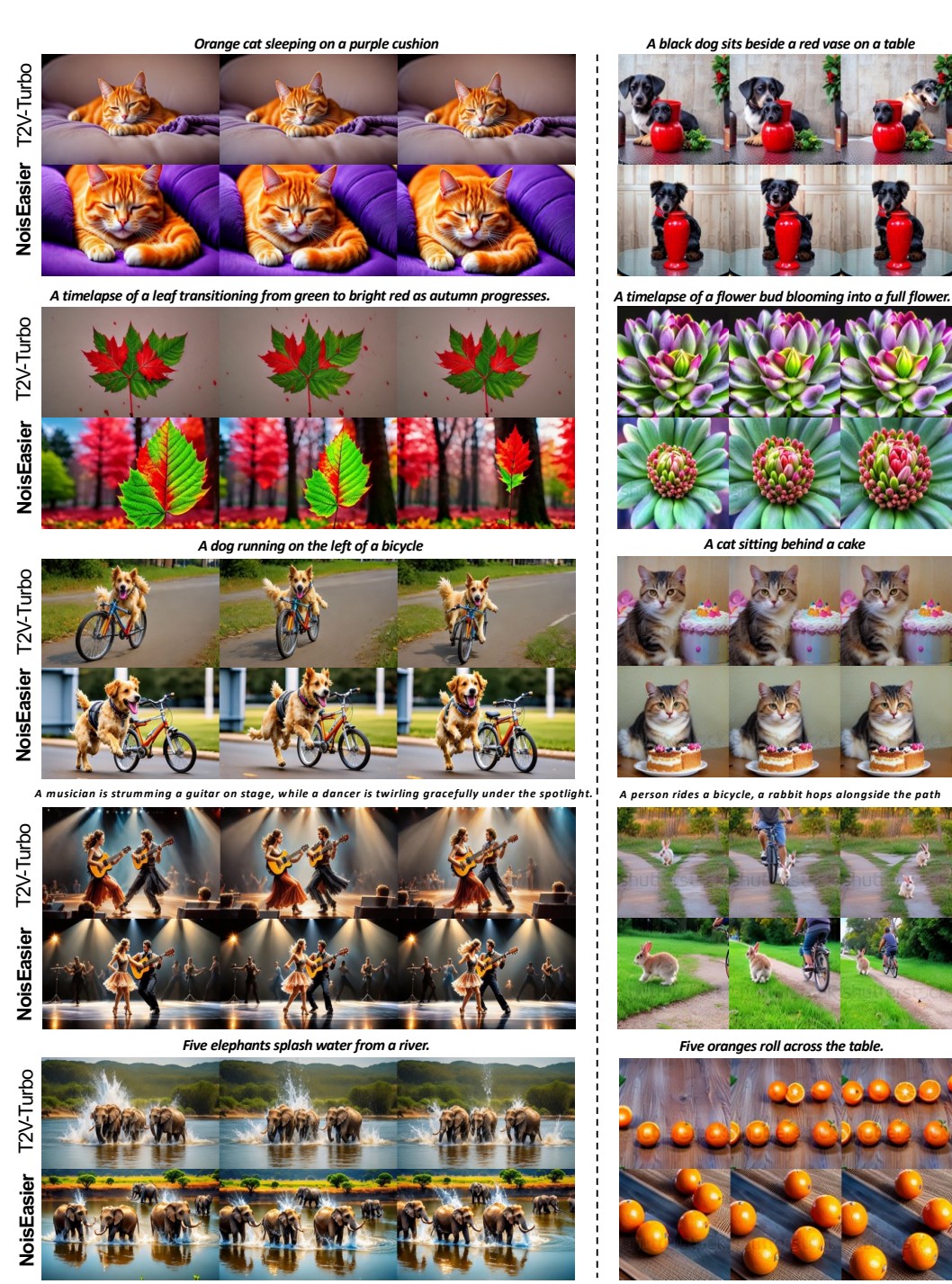

Figure 12: Qualitative results with and without NoisEasier on T2V-CompBench. **Left**: T2V-Turbo (VC2), 4-step generation; **Right**: T2V-Turbo (MS), 4-step generation.

