# OpenReview forum: "Make it NoisEasier: Boosting Text-to-Video Generation with Direct Noise Optimization"
_ICLR.cc/2026/Conference — ICLR 2026 Conference Withdrawn Submission_

### Official Review · Reviewer_Fu3g · 2025-10-18

**Soundness:** 2
**Presentation:** 3
**Contribution:** 2
**Rating:** 4
**Confidence:** 3

**Summary:**

The authors propose NoisEasier, a test-time training framework for optimizing video generation results.
NoisEasier optimizes latent noise directly using differentiable rewards during inference, leveraging fast video consistency models (e.g., T2V-Turbo) to enable full gradient backpropagation. It integrates multiple reward objectives to balance semantic alignment and motion quality, and introduces a negative-aware reward calibration strategy using LLM-generated distractors for fine-grained compositional feedback.
Experiments on VBench and T2V-CompBench show that NoisEasier consistently improves baselines.

**Strengths:**

1. Unlike traditional reward-based fine-tuning that updates model parameters (computationally expensive and inflexible) or fixed-model inference (poor adaptability), NoisEasier optimizes latent noise directly during inference. This avoids redundant model parameter updates and can dynamically adapt to different text prompts.

**Weaknesses:**

1. The paper assigns weights of 2 to HPSv2 and ViCLIP, and 1 to ImageReward and motion reward. What is the theoretical basis for this weight setting? Has a sensitivity analysis been conducted (e.g., testing weights such as 0.5, 1, 2, 3)
2. Has the framework been validated on other video consistency models (e.g., VideoLCM), and what is its adaptability to different backbones?
3 Negative samples are generated using GPT-4o, a closed-source LLM. If researchers only have access to open-source LLMs (e.g., Llama 3, Qwen), can these models generate high-quality "hard negatives" (with appropriate semantic perturbations) to ensure the effect of reward calibration?
4. For closed-source models (e.g., Gen-3, Kling) marked with "*" (reproduction results), what are the specific hyperparameters used in the reproduction (e.g., denoising steps, text encoder weights)? For open-source models like CogVideoX-2B, were the official recommended optimization strategies (e.g., dynamic prompt enhancement) enabled during testing? In addition, what GPUs were used to test comparative models such as CogVideoX-2B and Wan2.1-1.3B? What is the memory usage of NoisEasier during operation?
5. he qualitative examples and test prompts in the paper (e.g., "a dog firefighter rescues kittens", "six robots dance rhythmically") are mostly "low-dynamic, multi-object static scenes". Has the paper tested high-dynamic scenes (e.g., "a basketball game with collisions") to verify the robustness of the motion reward?

**Questions:**

see weakness

---

### Official Review · Reviewer_mPiR · 2025-10-30

**Soundness:** 2
**Presentation:** 3
**Contribution:** 2
**Rating:** 4
**Confidence:** 4

**Summary:**

This paper proposes a test time training framework to enhance T2V generation. The primary contribution is a method that directly optimizes the latent noise during inference using a combination of differentiable rewards. This approach is made computationally efficient by leveraging fast video consistency models. The paper also introduces a "negative-aware reward calibration" strategy that uses generated  negative prompts for contrastive reward calculation.

**Strengths:**

- The paper is clearly written and easy to follow.

- The paper conducts extensive ablation studies to validate the effectiveness of each proposed component.

**Weaknesses:**

- The paper's contribution is somewhat incremental. It combines existing techniques, including mixed reward feedbacks (e.g. T2V-Turbo) and test time optimization of the input noise (e.g. Initno). The paper claims the "negative-aware reward calibration" as a major contribution, but its effectiveness appears to be limited according to Figure 6 (human preference rate 51.7 vs 48.3).

- The paper lacks a critical baseline that optimizes the entire model with the same mixed reward model. Though the base models (T2V-Turbo) have already utilized HPSv2 and ViCLIP for reward-based model distillation, it is difficult to directly assess the necessity of optimizing the noise, as the reward and inference setups are different.

- The paper relies on a distilled 4-step consistency model for effective test time training. However, the test time training makes the method much slower than the distilled model, diminishing the advantage of the consistency model. Figure 5 provides a Quality-Cost Pareto analysis trying to demonstrate the proposed model is more efficient than CogVideoX-2B, but this comparison to an arbitrary model does not sufficiently prove the method's efficiency. For example, a distilled version of CogVideoX-2B might be more effective than the proposed method.

**Questions:**

- In Table 1 (Quantitative Evaluation on VBench), some open-source models have better scores than leading commercial models. For example, the average score of Open-Sora 2.0 is higher than Sora, and CogVideoX-2B is very close to Kling. Is there any explanation for this?

- Directly optimizing with the reward gradient is usually considered highly susceptible to reward hacking. Did the authors observe related failure cases during experiments (beyond the ablations on single-reward optimization)?

---

### Official Review · Reviewer_Hbw5 · 2025-11-02

**Soundness:** 3
**Presentation:** 3
**Contribution:** 3
**Rating:** 6
**Confidence:** 2

**Summary:**

The paper proposes NoisEasier, a test-time training framework for improving text-to-video (T2V) diffusion models by optimizing latent noise directly using differentiable reward signals. Rather than fine-tuning model weights (as in reward-based RLHF or preference optimization), NoisEasier refines the input noise during inference, leveraging video consistency models that allow full gradient backpropagation in only 4–8 denoising steps. Empirical Results show that NoisEasier delivers consistent and significant improvements over strong baselines on VBench and T2V-CompBench, with high efficiency.

**Strengths:**

1. While noise optimization has been explored in text-to-image and motion domains, this work is the first to successfully adapt it to video generation.

2. The authors show significant performance improvement over the T2V-Turbo base model and other strong models.

3. The paper is well-written and easy to follow.

**Weaknesses:**

1. Although results are strong on VBench and T2V-CompBench, both benchmarks focus primarily on short, synthetically structured prompts. The method’s effectiveness on longer, narrative-style prompts remains untested. More realistic evaluation is need to understand the generalizability of the method.

2. The framework heavily relies on existing image- and video-based reward models, which were trained on specific distributions and may not generalize. This might contribute to my previous point.

**Questions:**

See weeknesses.

---

### Note · Authors · 2025-11-14

I have read and agree with the venue's withdrawal policy on behalf of myself and my co-authors.